# N6-Methyladenosine Modification in the Metabolic Dysfunction-Associated Steatotic Liver Disease

**DOI:** 10.3390/nu17071158

**Published:** 2025-03-27

**Authors:** Satoru Matsuda, Moeka Nakashima, Akari Fukumoto, Naoko Suga

**Affiliations:** Department of Food Science and Nutrition, Nara Women’s University, Kita-Uoya Nishimachi, Nara 630-8506, Japan

**Keywords:** reactive oxygen species, N6-methyladenine, non-coding RNA, RNA binding protein, autophagy, mitophagy, liver dysfunction, MASLD

## Abstract

Epigenetics of N6-methyladenine (m6A) modification may play a key role during the regulation of various diseases, including metabolic dysfunction-associated steatotic liver disease (MASLD). The m6A modification has been shown to be accomplished via the exploitation of several players such as methyltransferases, demethylases, and/or methylation-binding molecules. Significantly, the m6A methylation can regulate the key eukaryotic transcriptome by affecting the subcellular localization, splicing, export, stability, translation, and decay of those RNAs. An increasing amount of data has designated that the m6A modification of RNAs can also modulate the expression of autophagy-related genes, which could also control the autophagy in hepatocytes. Oxidative stress with reactive oxygen species (ROS) can induce m6A RNA methylation, which might be associated with the regulation of mitochondrial autophagy (mitophagy) and/or the development of MASLD. Therefore, both autophagy and the m6A modification could play important roles in regulating the pathogenesis of MASLD. Comprehending the relationship between m6A and mitophagy may be helpful for the development of future therapeutic strategies against MASLD. This review would advance the understanding of the regulatory mechanisms of m6A RNA modification, focusing on the impact of mitochondrial dysregulation and mitophagy in the liver with MASLD.

## 1. Introduction

Metabolic dysfunction-associated steatotic liver disease (MASLD), formerly called non-alcoholic fatty liver disease (NAFLD), is well-defined based on the co-occurrence of hepatic steatosis with other metabolic diseases such as obesity, insulin resistance, and type 2 diabetes mellitus [1]. NAFLD has been retitled to MASLD, which more precisely reveals the recent comprehension of the disease [2]. It has been documented that MASLD is the most frequent chronic liver disease worldwide [2]. Yet, its detailed pathogenesis requires a more profound investigation. With the advancement of the disease, a fraction of MASLD individuals may finally progress to metabolic dysfunction-associated steatohepatitis (MASH), fatty liver fibrosis, cirrhosis, liver dysfunction, and/or hepatocellular carcinoma (HCC) [3]. In general, the MASLD includes an array of diseases from fatty liver disease with no apparent inflammatory appearances to MASH. This type of hepatic steatosis is frequently linked to pericyte fibrosis, lobular inflammation, and/or the apoptosis of hepatocytes, which cannot histologically separate from alcoholic steatohepatitis [4]. There is almost no actual therapeutic medicine for the comprehensive management of MASLD, although several lifestyle modifications including well-adjusted diets and/or weight loss diets could improve symptoms to some extent [5]. Now, MASLD has resulted in an enormous medical burden [6]. Consequently, the discovery of an effective treatment is urgent.

Epigenetic modifications usually involve nucleotide methylation, histone modification, and/or chromosomal remodeling [7], which may suggest new insights into the pathophysiology of various types of disease including MASLD [8]. In addition, inflammation, oxidative stress, and/or mitochondrial damage are commonly involved in various liver diseases [9,10]. RNA and DNA methylation are two of the most crucial epigenetic changes contributing to the progression of MASLD [9]. There are more than hundreds of chemical modifications of various types of RNA [11]. N6-methyladenosine (m6A) is a post-transcriptional RNA modification in eukaryotes, which can control the expression of several key genes [12,13]. Inside the construction of total RNAs, m6A is predominantly distributed in the protein coding sequence of messenger RNA (mRNA), 3′ untranslated region (UTR) of the mRNA, and/or regions nearby the stop codon [14]. A function of the m6A modification in the regulation of gene expression might be related to the pathological progressions of various disorders including metabolic, neurodegenerative, and/or cancerous diseases. In addition to the transcriptional regulation of encoding mRNAs, the m6A modification could also control the transcription of a variety of non-coding RNAs (ncRNAs) such as microRNA (miRNAs), long non-coding RNAs (lncRNAs), and/or circular RNAs (circRNAs) [15]. Hence, the m6A modification can influence the various types of cellular function including stress responses, proliferation, differentiation, organ development, and/or certain pathogenesis [16,17]. In general, the motif of m6A may differ among normal tissues and/or diseases, which may also suggest a role of m6A modification in human evolutionary adaptation and disease susceptibility [18]. Interestingly, the m6A-modified ncRNAs can also control the expression of m6A-related molecules, suggesting that these ncRNAs could be applied as underlying markers for the diagnostic or therapeutic tools of several diseases, including MASLD [19] (Figure 1).

Several studies have revealed the effect of m6A modification involved in some mechanisms of autophagy development, suggesting that the m6A modification can play a vital role in the regulation of autophagy [20,21]. For example, the m6A modification can stimulate inhibitory moments during the development of various types of autophagy [21]. Furthermore, the m6A modification could also disturb the formation of autophagosomes within cells [22]. Occasionally, the m6A modification could even endorse autophagy [23]. Evidence has described that the m6A modification may be a widespread incidence under oxidative stress. Roles of the m6A-related molecules have been implicated in the development of MASLD, which may contribute to the development of the superior treatment of MASLD. In this review, we summarize the up-to-date studies explaining the effect of the m6A modification in the pathogenesis of MASLD through the regulation of autophagy. Several roles of the m6A modification–autophagy axis has been deliberated for the favorable treatment of MASLD.

## 2. ROS, Inflammation, and the m6A-Modified RNAs Involved in MASLD

m6A is the most prevalent internal modification occurring in eukaryotic RNAs, affecting various biological processes of RNAs including splicing, translation, transport, and/or degradation [24,25]. m6A might be regulated by several enzymes such as methyltransferases and demethylases [26]. Methyltransferases, also named “writers”, may contain methyltransferase-like 14 (METTL14), methyltransferase-like 3 (METTL3), and Wilms tumor 1-associated protein (WTAP) with the function of methyl installation to the target adenosine [27]. The m6A modification can be removed by demethylases, also recognized as “erasers”, such as FTO and/or ALKBH5. Interestingly, the silencing of FTO can repress the invasion and/or proliferation of cancer cells via the m6A alteration of the myelocytomatosis oncogene (Myc) [28]. Silencing of the FTO might also impede IL-1β expression through the modification of signaling related to the nuclear factor κB (NF-κB) signaling pathway [29]. Additionally, several RNA-binding proteins, so-called “readers”, may contain nuclear IGF2BP1/2/3, HNRNPC, YTHDF1/2/3, YTHDC1, and cytoplasmic YTHDC2. These reader molecules could distinguish the m6A motif, achieving the functional modification of the m6A-RNAs [30].

The m6A-RNAs can play a crucial role in the inflammatory response induced by various oxidative stresses. It has been described that certain immunological stress triggered by lipopolysaccharide (LPS) stimulation could influence the level of m6A-RNAs [31]. Consequently, the downregulation of METTL3 could invert the LPS-provoked cell damage by increasing mRNA stability [32]. Also, the deletion of METTL3 might increase the expression of another gene, which could inhibit the NF-κB signaling in cells treated with LPSs [33]. Moreover, the altered expression of “reader” molecules may be associated with the development of various inflammation processes [34], which may also play imperative roles in several pathological progressions, including inflammatory and/or autoimmune responses [27,35]. Silencing the function of FTO could also prevent IL-1β expression via the NF-κB signaling pathway [29,36]. In addition, a decrease in YTHDF2 might initiate several inflammation reactions in hepatocellular carcinoma (HCC) [37]. Remarkably, the METTL3 expression may be upregulated through the increased production of ROS during inflammation [38]. Therefore, the ROS content may be increased with the related elevation of m6A levels in liver inflammation [25]. Interestingly, the methylation of m6A in several RNAs may be altered under oxidative stress [25]. The ROS level may increase the expression of YTHDF2 for the rise in m6A [25]. Hence, the modification of m6A may be a widespread event under the situation of oxidative stress. In addition, several immunological stress could also influence the m6A level of mRNAs [31,39]. In particular, the change in ROS levels with immunological stress might simultaneously contribute to the elevation of the m6A-RNA level [40]. Recently, it has been reported that METTL3 can promote the progression of MASLD by mediating the m6A methylation of fatty acid synthase mRNA [41]. Again, the m6A methylation may be closely associated with the increased level of ROS production, which might be related to the development of MASLD (Figure 2).

## 3. Mitochondrial Dysfunction and the Development of MASLD

Mitochondrial dysfunction may play an indispensable role in the development of MASLD, which is also an imperative cause of simple hepatocyte damage in liver toxicity [42]. In addition, unusual mitochondrial respiration and ATP superfluous production are prominent reasons for post-transplantation liver failure, in which mitochondrial dysfunction has been frequently observed [43]. Therefore, improving mitochondrial activity might contribute to developing tactics for the protection of hepatocytes [43]. In general, mitochondria are the main source of ROS both in physiological and under pathological conditions [43,44]. This step might also induce compromised lipid metabolism to exacerbate the development of MASLD [44]. However, the biological comprehensive function of the m6A modification in the mitochondrial dysfunction during the pathogenesis of MASLD remains unidentified. Interestingly, the m6A modification-related genes seem to be dispersed in intracellular signaling pathways possibly linked to NF-κB inflammation. In addition, the m6A-methylated transcripts can be augmented in pathways connected to several inflammatory immune responses [45]. Therefore, the mechanism by which mitochondria could organize the liver function, metabolism, and/or inflammation has rigorously been explored. Several studies have revealed that some factors such as oxidative stress and/or autophagy/mitophagy (a selective autophagy for damaged mitochondria) could affect mitochondrial function to control liver injury. In addition, mitochondrial dysfunction may be connected to the induction of excessive oxidative stress, which might cause the pathogenesis of several types of liver damage. Remarkably, it has been shown that the FTO protein could improve mitochondrial quality, which consequently regulates the oxidative stress in liver damage [46]. The FTO-dependent hepatic m6A methylation in several RNAs has been shown to be functionally important for liver protection, which may also suggest that FTO could be a potential therapeutic target for amending liver damage. Interestingly, a preceding study has discovered that the natural product rhein could bind to the active site of FTO [47], which may also improve lung injury induced from respiratory viruses [48]. It is possible that the m6A modification could play a significant role in the onset and/or progress of various disorders. It would be a beneficial challenge for applying the m6A technique to the treatment of human diseases including MASLD.

In several patients with MASLD, the disruption of lipid metabolisms mediated by mitochondrial dysfunction could lead to an extreme accumulation of triglycerides in many hepatocytes, which might induce hepatic steatosis [49]. These atypical mitochondria could be recognized by the reduced activity of respiratory chain enzymes. As autophagy/mitophagy could retain cellular homeostasis by eliminating nonfunctional and/or damaged molecules/mitochondria from cells, the suitable function of autophagy/mitophagy may be of great importance for protection. Low levels of β-oxidation and/or induced mild lipogenesis may induce lipid accumulation in hepatocytes, which might contribute to the additional production of ROS, causing hepatic inflammation and liver fibrosis [50]. Mitophagy could suppress the accumulation of dysfunctional mitochondria, superfluous oxidative stress, and/or severe inflammation. Furthermore, the transition from MASLD to NASH is not only based on steatosis but is also characterized by mitochondrial dysfunction [51]. Although mitochondria could reduce toxic offenses in the initial stage of liver diseases, prolonged uncontrolled ROS production by impaired mitochondria might result in severe damage to hepatocytes [52]. Therefore, upgrading mitochondria could be a potential strategy to prevent the progression of MASLD. For example, strong antioxidants might have potential to reverse mitochondrial dysfunction for clinical application against several liver diseases. It may also be critical to elucidate its regulatory mechanisms with antioxidants for therapeutic tactics for MASLD (Figure 3). Interestingly, it has been shown that mega-mitochondria or giant mitochondria are considered to be a pathological hallmark of the fate of liver parenchymal cells that leads to liver deterioration and eventually results in liver failure [53]. However, the cause and potential role of mega-mitochondria remain mostly unexplored.

## 4. m6A RNA Modification and Autophagy/Mitophagy in MASLD

Autophagy/mitophagy might tolerate cells through inhibiting oxidative stress by recycling their damaged cellular organelles, proteins, lipids, and other cellular components to maintain cellular homeostasis. Certain epigenetic modifications such as DNA methylations, histone modifications, and/or m6A RNAs methylation could thoroughly regulate the expression of key genes involved in this autophagy/mitophagy control. In particular, it is remarkable that the post-transcriptional regulation of ULK1 could be changed by m6A RNA modification, resulting in considerable inhibition of autophagy/mitophagy [54]. ULK1 is a key molecule for autophagy/mitophagy control. The m6A RNA sequencing and/or m6A qRT-PCR technologies have been used in studies of autophagy/mitophagy [54]. Additionally, the METTL3 protein may be an important factor connected to the modification of m6A for inhibitory effects [55]. Several investigations have confirmed that knocking down METTL3 could improve the symptoms of mouse model colitis [55], in which autophagy/mitophagy may be frequently stimulated in cells of the colitis [23,56]. Consequently, the comprehension of the m6A methylation mechanism for the regulation of autophagy might be indispensable for the development of therapeutic strategies [22,57]. Likewise, YTHDF1 might increase the translation of ATG2A and/or ATG14 autophagy-related genes by binding to the m6A-modified mRNAs of ATG2A and/or ATG14, which might subsequently support autophagy/mitophagy activity [58]. YTHDF1 may also assist in the translation of the other m6A-modified mRNAs [58]. Hence, knocking down YTHDF1 might reduce the ability of RNA-binding proteins to distinguish the site of m6A, inhibiting the translation of mRNAs, thus disturbing downstream molecular functions. The lncRNA of ZFAS1 could also control the expression of ATG10, which might control autophagy/mitophagy by impeding the PI3K/AKT signaling pathway [59]. Interestingly, YTHDF2 could hold eIF4G1 transcripts with m6A methylation, which might provoke the degradation of mRNAs, thereby considerably encouraging autophagy [60]. In addition, the m6A modification might also increase the stability of ZFAS1 RNAs [59]. Today, a number of studies have approved that the m6A modification might regulate the activation of autophagy by modulating the expression of ATG5, ULK1, YTHDF1, YTHDF2, and/or ZFAS1 [61]. These molecules have been recognized to be involved in the autophagy mechanism [22]. Elevated m6A modifications could accelerate the formation of autophagosomes, as well as the function of lysosomes [62,63]. Therefore, the m6A modification can be linked to the development of MASLD, wherein the m6A–autophagy axis plays critical roles (Figure 3).

By augmenting autophagy, the cellular regeneration of hepatocytes could be prominently restored [64]. The hepatocyte regeneration from acute liver injury may be beneficial, which has been characterized by a well-arranged regenerative process in the liver [65]. On the other hand, abnormal autophagy may result in the development of a variety of liver diseases, including MASLD with impaired liver regeneration [66]. In particular, atypical autophagic function and activated inflammation may be typical features in a damaged liver [66]. In fact, autophagy could be repressed by noticeable decreases in the expression of ULK1 and the LC3II/LC3I ratio in human MASLD patients [66]. Defective autophagy might be related to the expansion of resident Gram-negative bacteria in the intestine [67], which can suggest the relationship between the gut and liver via the alteration of autophagy [68]. The liver and intestinal tract are closely and functionally related [69]. In healthy conditions, the gut microbiota maintains ordinary function for the host by delivering necessary nutrients from dietary foods. The composition of gut the microbiota is flexible and dynamic, which may be related to diet, age, stress, and/or drugs. When the gut microbiota is under dysregulation, it could result in the initiation and progression of various liver diseases including MASLD, modulating immune responses. In MASLD patients, it has been observed that gut bacteria could transfer across the portal vein into the liver and trigger unusual activation of the immune system, leading to inflammation responses and liver injury [70]. Communication between the intestine and liver may be bidirectional. For example, hepatogenic inflammatory cytokines could damage the gut mucosal barrier function, which could also form a liver–gut vicious cycle during the pathology of MASLD [71]. Interestingly, it has been described that the m6A modification not only regulates intestinal mucosal immunity and intestinal barrier function but could also influence apoptosis and/or autophagy in gut cells [72]. Understanding the detailed interaction between m6A methylation and autophagy would be a striking topic in cellular biology. Therefore, further in-depth investigations are necessary to elucidate the mechanism for the modification of autophagy under pathological conditions.

## 5. Possible Molecular Mechanisms for the Development of MASLD

The liver is rich in immune cells including neutrophils, macrophages, and/or various lymphocytes. Therefore, the liver is thought of as an essential immune organ with a vital role in the host defense system against bacteria and/or their toxic products, such as lipopolysaccharides (LPSs). In fact, hepatocytes could make critical proteins and complements during several bacterial infections [73]. In addition, Kupffer cells residing within the liver could be activated to produce several cytokines [73]. As a main component of Gram-negative bacteria, the LPSs may be recognized by receptors on Kupffer cells, which could trigger the signal transduction pathway for producing kinds of inflammatory cytokines including IL-1β and/or TNF-α [74,75]. In particular, the gut-derived LPSs might provide a signal to initiate inflammasome formation [74,75]. However, excess production of the inflammatory cytokines might lead to several abnormal liver lesions such as a hepatic lobule structure [76]. Interestingly, the ROS level may be considerably increased with the simultaneous increase in the m6A methylation in the liver after LPS treatment [77]. Consistently, total m6A levels in the liver might be increased when treated with LPSs [45,78]. It has been suggested that ROS could control key epigenetic processes [79]. However, specific mechanisms are still mostly unclear in the modification of post-transcriptional gene expression mediated by m6A methylation. Interestingly, the change in the m6A modification might be linked to the increase in ROS levels. For example, it has been revealed that ROS could indorse the transcription of hypoxia-inducible factor 1 alpha (HIF-1α) by activating the HIF-1α promoter [80]. As a transcription factor, HIF-1α could direct the transcriptional setting by regulating several genes’ expression, in which methionine adenosyltransferase 2A (MAT2A) may be involved in the induction of transcriptional control [81,82]. MAT2A can catalyze the generation of S-adenosylmethionine (SAM), which is an imperative donor for methylation reactions, in which METTL3 could contribute methylation to m6A [83,84]. Interestingly, the insulin stimulation could increase the production of MAT2A, WTAP, and SAM, which might enable the m6A methylation of several RNAs [85]. The expression of these molecules might be increased by LPS treatment with elevated m6A levels, which may be related to the alteration in ROS production and HIF-1α expression levels [40]. High ROS levels in the liver might be related to the high expression of HIF-1α, which can consequently control the MAT2A levels. The change in ROS, HIF-1α, and MAT2A can contribute to the modification of m6A RNA methylation, which could in part play an essential role in the stimulation of the nucleotide-binding oligomerization domain (NOD)/nuclear factor κB (NF-κB) signaling pathway [40]. The NOD-like receptor (NLR) family fundamentally include NOD1, NOD2, and NOD-like receptor, forming inflammasomes [86]. NOD1 could recognize bacterial products, activating the NF-κB and MAPK signaling pathways by elevating the expression of inflammatory cytokines [87]. NOD1 may also be expressed in various hepatic cells including hepatocytes, Kupffer cells, and/or neutrophils [88]. Hence, the overexpression of NOD1 can be associated with the liver damage induced by LPS treatment [40].

## 6. Roles of the Gut Microbiota in the Treatment of MASLD

The presence of certain microorganisms in the diet could occasionally induce oxidative stress and metabolism disorders, which may affect the constitution of the gut microbiota and/or the well-being of the host. As the most principal posttranscriptional modification of eukaryotic RNAs, m6A RNA modification could lead to the therapeutic mechanism for the improvement of healthcare in patients with various diseases. Interestingly, *L. plantarum* and *A. muciniphila* in the gut microbiota can influence the specific m6A modification in mice, which might highlight epitranscriptomic adjustment for therapeutics with commensal bacteria [89]. Consequently, the gut microbiota and RNA epigenetics may develop an intricate cross-regulatory network [90]. For example, the deletion of the *YTHDF1* gene could increase *A. muciniphila* colonization to enhance anti-inflammatory effects as a feedback system by promoting the expression of the *Foxp3* gene via the m6A modification [91]. By means of this model indication, it would be possible to treat MASLD with some probiotics and/or fecal microbiota transplantation (FMT) [92,93], as shown in Figure 3. In particular, suitable FMT has a qualifying effect on high-fat-diet-induced obesity, which may be a result of a prominent effect on microbial alteration by FMT [94]. Remarkably, the mitigating effect with FMT can modify the intestinal lipid metabolism and/or the m6A methylation levels to reduce the obesity level [94] (Figure 3). Additionally, we have previously proposed the “engram theory”, with the concept that the adjustment of the gut microbiota may become one of the favorable tactics for the treatment of MASLD [95,96]. Thus, a non-invasive modification of the gut–brain–immune axis with the adjustment of the gut microbiota seems to hold promise. Highlighting the newest m6A research results for superior therapy against MASLD is now more vital than ever. However, further in-depth investigations are obligatory to develop effective therapeutic tactics based on the interaction among m6A methylation, autophagy, the gut microbiota, and/or engrams so as to provide new methods for the treatment of MASLD.

## 7. Future Perspectives

As a lot of studies have emphasized the key role of the gut microbiota in the development of MASLD, the gut microbiota for MASLD diagnosis and/or treatment has received growing interest. In addition, clinical studies have revealed alterations of the gut microbiota in patients with various liver diseases, which may also suggest a clinical direction of application as a tentative non-invasive biomarker for prognosis of the related liver disease [97,98]. Given the inadequate strategies for the treatment of MASLD, many studies on the gut microbiota would provide some novel approaches as well as challenges in the therapeutic innovation of MASLD. Also, more patients with liver diseases would benefit from this development. As shown here, lipid accumulation in the liver during the progression of MASLD might be associated with mitochondrial damage and/or mitophagy within hepatocytes. Lipid accumulation may also affect fatty acid beta-oxidation with the decreased ATP production in mitochondria, as post-transcriptional m6A methylation can prevalently participate in the modulation of mitochondrial gene expression, which might change the development of MASLD [46,99,100]. However, few investigations have uncovered the precise role of m6A in various inflammatory damages. In particular, the underlying regulatory mechanisms in the liver remain mostly unexplored. It would be required to search for the precise mechanisms for the development of advanced treatments.

The discovery of m6A methylation has brought a new feature to the field of post-transcriptional gene expression [101,102]. While certain m6A-related regulators could serve as novel therapeutic strategies for MASLD, the systematic evaluation of m6A regulator-related alterations could set a critical foundation for understanding the detailed characteristics of MASLD. Therefore, it should be further explored. Interestingly, the link between m6A methylation and cellular aging/senescence can offer a novel therapeutic target with important medical implications against various age-related disorders, including MASLD [103]. It is also meaningful that some miRNAs and/or small non-coding RNAs (18–24 nucleotides) could epigenetically regulate autophagy in relation to the pathology of MASLD [104,105,106]. The deregulation of specific miRNAs might yield a distinct character for each disorder and probably lead to a specific diagnosis of the disorder. Therefore, it would be interesting to survey the effect of miRNA on autophagy in the development of MASLD. Again, targeting specific m6A regulators may offer potential therapeutic approaches [107]. Few studies have identified useful molecular inhibitors targeting m6A regulation, while gene therapies with genome-editing expertise for several hereditary disorders have engrossed attention.

## 8. Conclusions

ROS may bring m6A RNA methylation associated with the regulation of mitophagy, which is involved in the development of MASLD. Future studies such as RNA methylation sequencing in MASLD patients should provide evidence suggesting that the m6A-mediated regulation of ncRNAs could play a role in the progression of MAFLD, offering potential avenues for future diagnostic strategies. Comprehending the correlation between m6A RNA methylation and mitophagy is indispensable for the development of innovative therapeutic strategies against MASLD.

## Figures and Tables

**Figure 1 nutrients-17-01158-f001:**
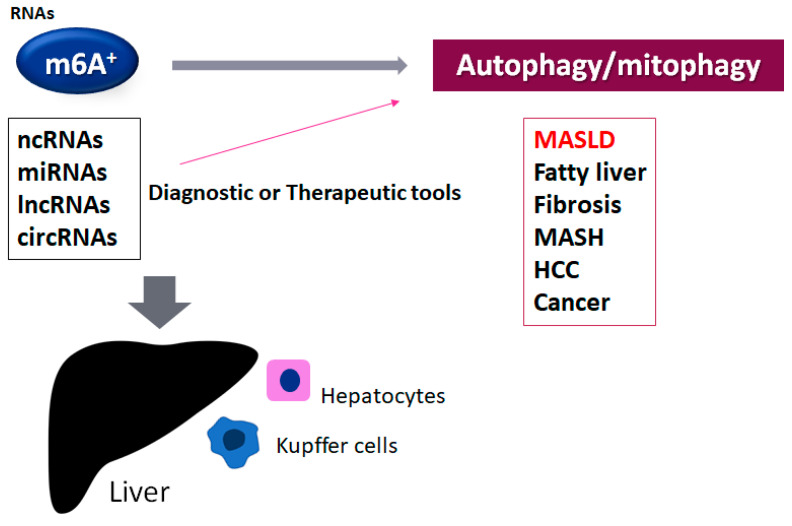
An illustration of the connection of the m6A-methylated RNAs to various liver disorders. The roles of the alteration of autophagy/mitophagy by the m6A-methylated ncRNAs such as miRNA, lncRNA, and circRNA have been suggested for liver dysfunction such as MASLD, fatty liver disease, liver fibrosis, metabolic dysfunction-associated steatohepatitis (MASH), hepatocellular carcinoma (HCC), and other cancers. The m6A-modified ncRNAs can be diagnostic and/or therapeutic tools for these disorders. The plus sign shows the existence of the modification of m6A in RNAs.

**Figure 2 nutrients-17-01158-f002:**
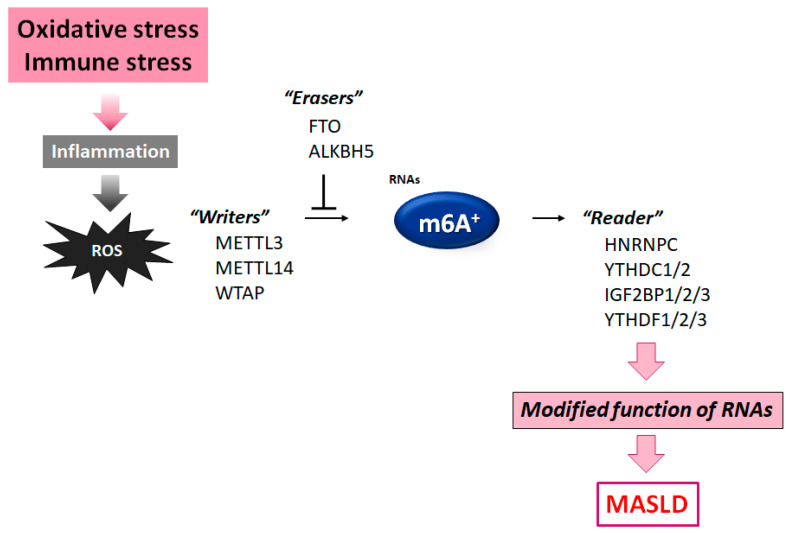
Diagram of the molecules comprising the m6A RNA methylation. The m6A modification is controlled by some methyltransferase “writers” and/or other demethylase “erasers”. Modified m6A-RNA binding proteins with the intracellular signaling function are called “readers”. Instance molecules are shown for each player. Immune stress and/or inflammation with ROS production may influence the function of these molecules. The arrowhead indicates stimulation, whereas the hammerhead denotes inhibition. Note that some critical pathways have been excluded for clarity.

**Figure 3 nutrients-17-01158-f003:**
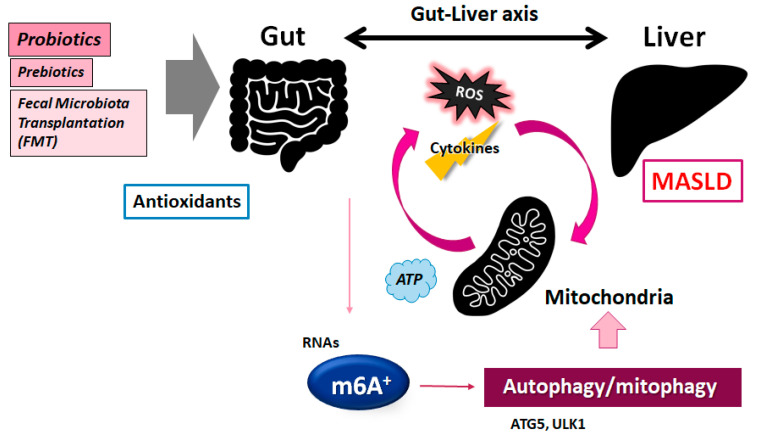
A graphic illustration of the possible tactics against the pathogenesis of MASLD. Some probiotics and/or fecal microbiota transplantation (FMT) may contribute to the modification of the gut microbiota in the host for the adjustment of autophagy/mitophagy as well as the m6A methylation of RNAs, which might be helpful for the treatment of MASLD. The arrowhead shows stimulation. Note that several significant functions such as cytokine induction and/or inflammatory intricate reactions have been lacking for clarity.

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
