# Peer review of "N6-Methyladenosine Modification in the Metabolic Dysfunction-Associated Steatotic Liver Disease"

_nutrients, 2025, doi:10.3390/nu17071158_

Round 1
Reviewer 1 Report
Comments and Suggestions for Authors
1. Several grammatical errors and awkward phrasings are present throughout the manuscript. For instance, in the abstract, the phrase "which might is associated with the regulation of mitochondrial autophagy (mitophagy) and/or the development of MASLD" should be corrected to "which might be associated with the regulation of mitochondrial autophagy (mitophagy) and/or the development of MASLD." A thorough proofreading by a native English speaker or language editing service is recommended.
2. The manuscript frequently uses "oxidative stresses", but in scientific writing, "oxidative stress" is generally treated as an uncountable noun. Consider revising for consistency.
3. The review highlights the importance of m6A modification in MASLD but lacks concrete experimental evidence linking these processes in the context of liver disease. If applicable, suggest experimental approaches such as in vitro models using hepatocytes with METTL3/METTL14 knockdowns or in vivo models with hepatic m6A modulation.
4. The manuscript frequently references pathways such as NF-κB and mitophagy but lacks a mechanistic explanation of how m6A directly influences these pathways.
5. The manuscript suggests that m6A-modified ncRNAs may serve as diagnostic or therapeutic tools for MASLD but does not specify how these biomarkers compare to existing clinical markers.
6. The manuscript alternates between "MASLD" and "NAFLD" in multiple places. Since the new nomenclature is MASLD, ensure consistency throughout the text.
7. The section discussing gut microbiota interactions with m6A modification is interesting but lacks references to studies showing direct evidence of this link in liver disease models.
8. The manuscript suggests that m6A modification could enhance mitophagy, but other studies have reported that excessive m6A methylation could impair autophagy. Addressing potential contradictions in the literature will improve the manuscript’s objectivity.
9. The conclusion suggests that future research should investigate m6A’s role in MASLD, but it does not provide specific experimental strategies. Suggesting studies such as RNA methylation sequencing in MASLD patients or mouse models will make this section more actionable.
Several grammatical errors and awkward phrasings are present throughout the manuscript. For instance, in the abstract, the phrase "which might is associated with the regulation of mitochondrial autophagy (mitophagy) and/or the development of MASLD" should be corrected to "which might be associated with the regulation of mitochondrial autophagy (mitophagy) and/or the development of MASLD." A thorough proofreading by a native English speaker or language editing service is recommended.
Author Response
For R1
- Several grammatical errors and awkward phrasings are present throughout the manuscript. For instance, in the abstract, the phrase "which might is associated with the regulation of mitochondrial autophagy (mitophagy) and/or the development of MASLD" should be corrected to "which might be associated with the regulation of mitochondrial autophagy (mitophagy) and/or the development of MASLD." A thorough proofreading by a native English speaker or language editing service is recommended.
Corrected. According to these suggestions, we have gone over the text/abstract and amended typos and grammatical errors as much as possible to improve the manuscript more helpful to the readers.
- The manuscript frequently uses "oxidative stresses", but in scientific writing, "oxidative stress" is generally treated as an uncountable noun. Consider revising for consistency.
Six times of "oxidative stresses" use have been corrected to "oxidative stress". Thank you so much.
- The review highlights the importance of m6A modification in MASLD but lacks concrete experimental evidence linking these processes in the context of liver disease. If applicable, suggest experimental approaches such as in vitro models using hepatocytes with METTL3/METTL14 knockdowns or in vivo models with hepatic m6A modulation.
Recently, it has been reported that METTL3 can promote the progression of MASLD by mediating m6A methylation of fatty acid synthase mRNA. In addition, the m6A RNA methylation and implications for hepatic lipid metabolism has been also described. We have mentioned these in the text.
Ref-41. Li Q, Xiang J. METTL3 promotes the progression of non-alcoholic fatty liver disease by mediating m6A methylation of FAS. Sci Rep. 2025 Feb 20;15(1):6162.
Ref-99. Ming X, Chen S, Li H, Wang Y, Zhou L, Lv Y. m6A RNA Methylation and Implications for Hepatic Lipid Metabolism. DNA Cell Biol. 2024, 43(6), 271-278.
- The manuscript frequently references pathways such as NF-κB and mitophagy but lacks a mechanistic explanation of how m6A directly influences these pathways.
For example, we have mentioned this in the second section, as followings.
The m6A modification can be removed by demethylases, also recognized as “erasers” such as FTO. Silencing of the FTO might impede the IL-1β expression through the modification of signaling related to the nuclear factor κB (NF-κB) signaling pathway [29].
- The manuscript suggests that m6A-modified ncRNAs may serve as diagnostic or therapeutic tools for MASLD but does not specify how these biomarkers compare to existing clinical markers.
Now, we have provided the concept for the application of m6A-modified ncRNAs in this manuscript. However, comparing the m6A-modified ncRNAs to existing clinical markers would be the most important issue in the future clinical studies.
- The manuscript alternates between "MASLD" and "NAFLD" in multiple places. Since the new nomenclature is MASLD, ensure consistency throughout the text.
"MASLD" has been used throughout the text. Thank you so much.
- The section discussing gut microbiota interactions with m6A modification is interesting but lacks references to studies showing direct evidence of this link in liver disease models.
At present, there are scarce studies showing the direct evidence of the link between gut microbiota interactions with m6A modification and liver disease models. We hope that this paper would encourage several relevant researchers to study them.
- The manuscript suggests that m6A modification could enhance mitophagy, but other studies have reported that excessive m6A methylation could impair autophagy. Addressing potential contradictions in the literature will improve the manuscript’s objectivity.
This is a very difficult issue to explain in this manuscript. Our speculation is that appropriate m6A methylation might be good for autophagy, but excessive m6A. modification could impair autophagy. Although this is an important issue, we have no evidence about it.
- The conclusion suggests that future research should investigate m6A’s role in MASLD, but it does not provide specific experimental strategies. Suggesting studies such as RNA methylation sequencing in MASLD patients or mouse models will make this section more actionable.
Exactly. Therefore, we have mentioned the following sentence in the conclusion section. Future studies such as RNA methylation sequencing in MASLD patients should provide evidence suggesting that m6A-mediated regulation of ncRNAs could play a role in the progression of MAFLD, offering potential avenues for future diagnostic strategies.
Comments on the Quality of English Language
Several grammatical errors and awkward phrasings are present throughout the manuscript. For instance, in the abstract, the phrase "which might is associated with the regulation of mitochondrial autophagy (mitophagy) and/or the development of MASLD" should be corrected to "which might be associated with the regulation of mitochondrial autophagy (mitophagy) and/or the development of MASLD." A thorough proofreading by a native English speaker or language editing service is recommended.
According to these suggestions, we have again gone over the text/abstract and amended typos and grammatical errors as much as possible to improve the manuscript more helpful to the readers.
Reviewer 2 Report
Comments and Suggestions for Authors
In the current review, the authors presented studies wich explain the effect of the N6‐methyladenine modification in the pathogenesis of metabolic dysfunction-associated steatotic liver disease through the regulation of autophagy. Several roles of the m6A modification-autophagy axis has been deliberated for the favorable treatment of MASLD.
Some suggestions:
1.Add please in which databases did you search the articles.
2.Line 32: add please example of “metabolic disease”.
3.Line 33-34, You wrote: “MASLD is the most frequent chronic liver disease in global [2]”. Give please some statistical data.
4.line 41-42, you wrote: “There is almost no actual therapeutic medicine for the comprehensive management of MASLD, although..” Nowadays there are several therapeutic approaches besides diet and lifestyle. You can list them.
5.Lines 64-65, give please more details concerning the statement “In general, the motif of m6A may differ among normal tissues and/or diseases [18]”
6.Point 4. m6A RNA modification and autophagy/mitophagy in MASLD. Several examples are given. In my opinion these should be more detailed.
The present review is of interest and well-organized. Developing a new treatment for MASLD represents a future subject. The paper should be of interest to scientists working in the field of MASLD as well as others with closely related research interests. However, I consider this article is not suitable for publication in Nutrients. It would be more appropriate for Biomedicines, Medicina or IJMS.
Author Response
For R2
In the current review, the authors presented studies wich explain the effect of the N6‐methyladenine modification in the pathogenesis of metabolic dysfunction-associated steatotic liver disease through the regulation of autophagy. Several roles of the m6A modification-autophagy axis has been deliberated for the favorable treatment of MASLD.
Some suggestions:
1.Add please in which databases did you search the articles.
Mainly, it was pubmed central.
2.Line 32: add please example of “metabolic disease”.
We have mentioned this as following. MASLD is well-defined based on the co-occurrence of hepatic steatosis with other metabolic diseases such as obesity, insulin resistance, and type 2 diabetes mellitus
3.Line 33-34, You wrote: “MASLD is the most frequent chronic liver disease in global [2]”. Give please some statistical data.
We have no direct statistical data, but citing the reference-2. In addition, the sentence has been amended.
4.line 41-42, you wrote: “There is almost no actual therapeutic medicine for the comprehensive management of MASLD, although..” Nowadays there are several therapeutic approaches besides diet and lifestyle. You can list them.
We cannot list them at present. There are scarce literatures showing the direct evidence.
5.Lines 64-65, give please more details concerning the statement “In general, the motif of m6A may differ among normal tissues and/or diseases [18]”
The sentence has been amended, accordingly.
6.Point 4. m6A RNA modification and autophagy/mitophagy in MASLD. Several examples are given. In my opinion these should be more detailed.
The text of section 4 has been amended for being more detailed.
The present review is of interest and well-organized. Developing a new treatment for MASLD represents a future subject. The paper should be of interest to scientists working in the field of MASLD as well as others with closely related research interests. However, I consider this article is not suitable for publication in Nutrients. It would be more appropriate for Biomedicines, Medicina or IJMS.
Thank you so much for the good evaluation to our manuscript.
Round 2
Reviewer 1 Report
Comments and Suggestions for Authors
No more comments